# Separation of α-Lactalbumin Enriched Fraction from Bovine Native Whey Concentrate by Combining Membrane and High-Pressure Processing

**DOI:** 10.3390/foods12030480

**Published:** 2023-01-19

**Authors:** María Romo, Massimo Castellari, Dinar Fartdinov, Xavier Felipe

**Affiliations:** 1Food Processing and Engineering Programme, Institute for Food and Agricultural Research and Technology (IRTA), Granja Camps i Armet s/n, Monells, 17121 Girona, Spain; 2Food Safety and Functionality Programme, Institute for Food and Agricultural Research and Technology (IRTA), Granja Camps i Armet s/n, Monells, 17121 Girona, Spain

**Keywords:** HPP, β-Lactoglobulin, α-Lactalbumin, whey proteins, bovine concentrated whey

## Abstract

Whey exhibits interesting nutritional properties, but its high β-Lactoglobulin (β-Lg) content could be a concern in infant food applications. In this study, high-pressure processing (HPP) was assessed as a β-Lg removal strategy to generate an enriched α-Lactalbumin (α-La) fraction from bovine native whey concentrate. Different HPP treatment parameters were considered: initial pH (physiological and acidified), sample temperature (7–35 °C), pressure (0–600 MPa) and processing time (0–490 s). The conditions providing the best α-La yield and α-La purification degree balance (46.16% and 80.21%, respectively) were 4 min (600 MPa, 23 °C), despite the significant decrease of the surface hydrophobicity and the total thiol content indexes in the α-La-enriched fraction. Under our working conditions, the general effects of HPP on α-La and β-Lg agreed with results reported in other studies of cow milk or whey. Notwithstanding, our results also indicated that the use of native whey concentrate could improve the β-Lg precipitation degree and the α-La purification degree, in comparison to raw milk or whey. Future studies should include further characterization of the α-La-enriched fraction and the implementation of membrane concentration and HPP treatment to valorize cheese whey.

## 1. Introduction

Whey represents around 85–90% of the total milk volume, and is the most important coproduct of cheese elaboration; nearly 55 million tons of whey were generated in Europe in 2019 [1]. Whey has been traditionally used as animal feed, but it is now considered a valuable source of high-quality proteins—with interesting applications for human consumption [2]. Several studies underlined the high digestibility of whey proteins (WPs) and their antioxidant, antihypertensive, anticancer, antidiabetic, hypocholesterolemic and antimicrobial activities [3,4]. Therefore, multiple applications of WPs have been described in the literature, e.g., as edible coating and delivery system [5,6], as a bioactive peptides source [7,8] or as food additives [9,10].

Depending on the production process, whey can be classified as sweet (liquid fraction obtained after enzymatic coagulation of milk caseins), acid (from acid coagulation of milk caseins) or ideal whey (liquid fraction after removal of caseins and fat from milk using physical methods). Native whey has a particularity: as it does not suffer any fermentation or acidification, its proteins mainly maintain their native form [11,12].

Bovine whey contains between 18–25% of the total milk proteins, namely β-Lactoglobulin (β-Lg; 18.4 KDa) and α-Lactalbumin (α-La; 14.2 KDa), the most relevant WPs. In cow milk, β-Lg and α-La account for 58% and 20%, respectively, of the total content of WPs [13].

Different β-Lg isoforms can be found in cow milk, with A and B the most predominant [14]. β-Lg conformation also depends on the pH of the medium; under physiological conditions, the protein forms a dimeric structure (which dissociates into monomers at extreme pHs: <3 and >9), while, between 3.5 and 5.5 pH, the isoform A participates in the formation of octamers [15]. Additionally, it should be noted that β-Lg, which is absent in human milk, has been related to infant allergies when bovine milk or WP concentrates (WPC) and isolates (WPI) are used in infant food formulations [16,17].

As such, there has been significant recent interest, within the dairy industry, in the development of bovine WPC and WPI with a more favorable ratio between α-La and β-Lg [18].

Several approaches to increase the ratio α-La/β-Lg in whey have been described in the literature, such as: (i) chromatography [19,20], (ii) selective precipitation [21,22], (iii) hydrolysis [23,24] and (iv) filtration [25,26]. However, these techniques could show some limitations when implemented at the industrial scale, such as low selectivity, use of additives, long processing time and high economical cost [27].

High-pressure processing (HPP) is a non-thermal food preservation technology which is mainly employed to extend the shelf-life of a large variety of food products with minimal impact on their organoleptic and nutritional values [28,29]. Notwithstanding, HPP treatment usually influences the protein structure, provoking conformational changes or a partial denaturation that can affect functionality [30].

Studies carried out on raw milk showed that bovine β-Lg was partially denatured and dissociated into native monomers at pressures higher than 100 MPa. Exposure to pressures higher than 300 MPa induced irreversible modifications in its tertiary structure, leading to aggregate formation and compromising its native conformation, while α-La was not affected by pressures up to 500 MPa [31,32].

The high susceptibility of β-Lg to HPP has been related to the free cysteine residue in its molecule, which can promote the aggregation and precipitation of this protein when HPP is applied by forming a disulfide bridge with another β-Lg molecule [33,34].

Working on the obtention of an α-La-enriched fraction from whey, Huppertz et al. [35] observed higher percentages of denaturation for β-Lg than for α-La in cow cheese whey treated at 600 MPa for 30 min. An α-La-enriched fraction from acidified raw cheese whey treated with HPP (600 MPa, 15 min) was also obtained by Marciniak et al. [36], while Tohuami et al. [37] combined HPP (600 MPa, 5 min) with ultrafiltration to separate α-La from pasteurized skimmed milk for a fraction with an α-La purity of 60.84%. On the other hand, ultrafiltration, based on membranes with pore size ranging between 1 nm to 500 nm, has been widely applied in the dairy industry to concentrate milk, among other applications [38].

In these studies, HPP was applied to products already exposed to high temperature or fermentation processes which could have induced a partial denaturation of the whey proteins before the high-pressure treatment [39,40]. For this reason, to the best of our knowledge, there is a lack of knowledge about the effects of HPP on native whey proteins.

Thus, the aim of this work was to study the effects of HPP treatments on main proteins (α-La and β-Lg) in bovine native whey concentrate obtained from fresh milk, by ultrafiltration at pilot plant level, without any heat treatment. Concentration of native whey was regarded as a suitable solution in future industrial applications to obtain high value α-La-enriched fractions while overcoming the HPP process limitations in terms of working capacity (L/h) [41].

Different combinations of pH, and the initial temperature of the sample—as well as of pressure and HPP processing time—were assessed, in order to obtain a suitable α-La-enriched fraction.

## 2. Materials and Methods

### 2.1. Milk and Whey Samples

Five commercial batches (500 L each) of raw skimmed cow milk were provided by a local farm (Ruidellots de la Selva, Girona, Spain). All processes and analyses were carried out at IRTA’s facilities, where data were collected. All membrane processes were performed according to fabricant specifications.

Each milk batch was first microfiltered (SW40 MMS AG Membrane Systems, Urdof, Switzerland) with a TAMI ISOFLUX ceramic membrane (1.4 μm, 1.05 m^2^ TAMI Industries, 26111 Nyons Cedex, France) for microorganism removal. Flowrate (cross-flow permeate) was set at 425 Lm^−2^h^−1^ and temperature at 50 °C, while transmembrane pressure was maintained at 1.2 bar. To separate the native whey, milk was further microfiltered with a Dairy-Pro polyethersulfone polymeric membrane (0.1 μm; 4,.6 m^2^; 1.1 mm feed spacer; from KOCH Separation Solutions, Commonwealth, MA, USA). Flowrate was set at 20.6 Lm^−2^h^−1^ and temperature at 50 °C, while transmembrane pressure was maintained at 1.8 bar.

The native whey was then concentrated sequentially using the SW40 ultrafilter equipped with a Dairy-Pro polyethersulfone polymeric membrane (10 kDa; 6.1 m^2^; 1.1 mm feed spacer; from KOCH Separation Solutions, Commonwealth, MA, USA) and a SW18 filter (MMS AG Membrane Systems, Urdof, Switzerland) equipped with a Dairy-Pro polyethersulfone polymeric membrane (5 kDa; 0.32 m^2^; 0.86 mm feed spacer; SUEZ Water Technologies & Solutions, Trevose, PA, USA). Flowrate was set at 10.3 Lm^−2^h^−1^ and 21.42 Lm^−2^h^−1^, respectively. Temperature was maintained between 10 °C and 25 °C during the process, while transmembrane pressure was maintained at 1.8 and 3.0 (inlet pressure) bars, respectively. Ultrafiltration processes were carried out within 24 h, maintaining the product under refrigerated conditions (4 °C). Native whey concentrate (NWC) was added to 0.33% of sodium azide, stored refrigerated at 4 °C and HPP treated within 24 h. All the processed samples were kept at 4 °C and analyzed within 1 week. The proximate composition of the five concentrated whey batches can be shown in Table 1.

### 2.2. Chemicals and Standards

Acetonitrile HPLC grade, trifluoracetic acid (TFA), hydrochloric acid (HCl), 2-mercaptoethanol, sodium citrate, urea, 8-anilino-1-naphtalenesulphonic acid (ANS), (ethylenedinitrilo)tetraacetic acid (EDTA), dibasic potassic phosphate, 5,5′-dithio-bis-(2-nitrobenzoic) acid (DTNB), bovine serum albumin (BSA), DL-Dithiothreitol (DTT), 2,2-Bis(hydroxymethyl)-2,2′,2′’-nitrilotriethanol (Bis-Tris) and sodium azide were provided by Sigma Aldrich (Sigma-Aldrich Merck, Darmstadt, Germany). Reference standards of bovine whey proteins (α-La, β-Lg A and β-Lg B isoforms) were provided by Cerilliant (Sigma-Aldrich Merck, Darmstadt, Germany).

### 2.3. Proximate Composition Parameters

Fat, protein and dry matter content were determined following standard protocols, with minor modifications. Fat content was determined following the protocol ISO 1211/IDF1 [42]. Protein content was calculated according to the Kjeldahl method following the protocol ISO 8968-3/IDF20-3 [43]. Total dry matter was determined gravimetrically according to the protocol ISO 2920:2004/IDF58:2004 [44]. Ashes were estimated gravimetrically according to BOE-A-1977-16116 [45]. Lactose was calculated from the total dry matter by subtracting the total fat, total protein and ashes. The pH of NWCs was determined with a pH-meter (sesnION + PH3, HACH Co., Loveland, CO, USA).

### 2.4. High Pressure Processing

HPP experiments were carried out with a Wave6000/120 industrial equipment (Hiperbaric, Burgos, Spain), with water as the pressure transmission medium. The compression rate was 150 MPa/min, while the decompression was less than 2 s, according to the data obtained from the equipment SCADA software. Samples (50 mL) were placed in high-density polyethylene (HDPE) bottles (Nalgene, Thermo Fisher Scientific Inc., Waltham, MA, USA) and equilibrated at the target temperature before processing.

A first set of experiments was carried out with NWCs from the first milk batch to study the effects of main processing parameters on whey proteins, based on a central composite design (CCD) (orthogonal axial value off 1.287, 5 central points, 1 replicate) developed with the JMP13 statistic software (SAS Institute Inc., Cary, USA). Different levels of initial pH of the sample (physiological and 4.60), processing time (from 47 s to 433 s), initial sample temperature (from 7 °C to 39 °C) and pressure (from 145 MPa to 655 MPa) were combined. Acidification of the samples was carried out by adding aqueous HCl until pH 4.60 (dilution factor was taking into consideration for later calculations). Untreated samples, for both physiological (P-pH) and acidified (pH 4.6) conditions, were used as controls. The design consisted of 20 experimental combinations which were performed in random order (Table 2).

A second set of experiments was performed to study the effects of processing time. Samples of four NWCs (NWC2 to NCW5), both at physiological pH (P-pH) and acidified (pH 4.6), were treated at 600 MPa for 2, 4, 8, 10, 15 and 30 min at room temperature (23 °C). Acidification of the samples was carried out by adding aqueous HCl until pH 4.60. Untreated samples for both P-pH and acidified (pH 4.6) conditions were used as controls. Each one of the four NWC batches was HPP treated independently (one process replicate for each NWC and process time).

NWCs samples from both experiments, once HPP treated, were centrifuged (Beckman Avanti^®^ JXN-30, Beckman Coulter, Inc., Brea, CA, USA) at 3270× *g* (4 °C, 20 min). Supernatant (NWC_sup_) and pellets were weighed, and the volume of the supernatant was calculated considering its density.

All samples were properly codified and identified in order to compare the different analyzed parameters between them.

### 2.5. HPLC Quantification of the Proteins

HPLC quantification was performed according to Marciniak et al. [36], with some modifications. Analysis of whey proteins (α-La, β-Lg A and β-Lg B) was carried out with a binary HPLC pump 1525 equipped with a 717 plus Autosampler, a photodiode array detector 2996 (Waters, Milford, MA, USA) and an Aeris Widepore XB-C8 column (3.6 μm particle diameter, 150 × 4.6 mm) with a SecurityGuard Ultra cartridge (30 × 4.6 mm) of the same stationary phase (Phenomenex, Torrance, CA, USA). Column temperature was set at 35 °C and flowrate at 1 mL·min^−1^. Chromatographic separation was performed with a gradient elution between mobile phase A (trifluoroacetic acid: water, 1:999 *v*/*v*) and B (trifluoroacetic acid: acetonitrile, 1:999 *v*/*v*), by varying (linearly) the percentage of mobile phase B from initial 38% to 43% in 8 min, then to80 % in 12 min. The system was controlled by Empower 2 software (Waters, Milford, MA, USA).

NWC_sup_ samples were added with HCl 1 N to adjust the pH up to 4.6, diluted 1:1 (*v*/*v*) with buffer (0.1 M Bis-Tris, 0.3% 2-mercaptoethanol, 5.37 mM sodium citrate), and further diluted 1:500 (*v*/*v*) with mobile phase (62% A and 38% B). Diluted samples were filtered through a 0.45 μm cellulose acetate syringe filter (Scharlab, S.L., Barcelona, Spain) and injected (25 μL). The signal, at 214 nm, was used for quantitation purposes. Protein identification was performed by comparing retention time and ultraviolet (UV) spectra of the peaks with those of the pure compounds, while quantification was carried out by using external standard calibration curves, created by injecting known amounts of pure commercial standards and recording the absorbance at 214 nm.

### 2.6. Process Performance Parameters

Main process performance parameters were calculated for NWC_sup_ according to the following equations:

α-La yield (Y_α-La_):(1)$$Y_{\alpha -\mathrm{La}}\;\left(\%\right)=\frac{{\left[\alpha \mathrm{La}\right]}_{S}\;\cdot {\;V}_{S}}{{\left[\alpha \mathrm{La}\right]}_{C\;}\cdot {\;V}_{C}}\cdot 100$$

α-La Purification degree (Pur_α-La_):(2)$${\mathrm{Pur}}_{\alpha -\mathrm{La}}\;\left(\%\right)=\frac{{\left[\alpha \mathrm{La}\right]}_{S}\;}{[\alpha \mathrm{La}{]}_{S}+[\beta \mathrm{LgA}{]}_{S}+{\left[\beta \mathrm{LgB}\right]}_{S}}\cdot 100$$

β-Lg A precipitation degree (Pre_β-LgA_):(3)$${\mathrm{Pre}}_{\beta -\mathrm{LgA}}\;\left(\%\right)=\frac{{\left[\beta \mathrm{LgA}\right]}_{S}\;\cdot {\;V}_{S}}{{\left[\beta \mathrm{LgA}\right]}_{C\;}\cdot {\;V}_{C}}\cdot 100$$
and β-Lg B precipitation degree (Pre_β-LgB_):(4)$${\mathrm{Pre}}_{\beta -\mathrm{LgB}}\;\left(\%\right)=\frac{{\left[\beta \mathrm{LgB}\right]}_{S}\;\cdot {\;V}_{S}}{{\left[\beta \mathrm{LgB}\right]}_{C\;}\cdot {\;V}_{C}}\cdot 100$$
where:

[αLa]_s_ = concentration of α-La in NWC_sup_ after the HPP treatment

[αLa]_c_ = concentration of α-La in the corresponding control sample (no HPP, P-pH).

[βLgA]_s_ = concentration of β-Lg A in NWC_sup_ after the HPP treatment

[βLgA]_c_ = concentration of β-Lg A in the corresponding control sample (no HPP, P-pH).

[βLgB]_s_ = concentration of β-Lg B in NWC_sup_ after the HPP treatment

[βLgB]_c_ = concentration of β-Lg B in the corresponding control sample (no HPP, P-pH).

V_s_ = volume of NWC_sup_ recovered after the HPP treatment

V_c_ = volume of the sample before the HPP treatment

### 2.7. Hydrophobicity Index

The assessment of Hydrophobicity Index (%) was performed according to Steen et al. [46], with small adjustments. Samples were diluted (0.16, 0.08, 0.04, 0.02, 0.01, 0.005 and 0.0025 *v*/*v*%) in dibasic potassic phosphate buffer (0.05 M, pH 7.40). Then, 4 mL of the diluted samples was incubated with 20 μL of ANS 8 mM for 10 min in darkness and then fluorescence (λex = 390 nm, λem = 480 nm) before being measured in a Varioskan Flash (Thermo Fisher Scientific, Vantaa, Finland). Hydrophobicity Index was calculated as a percentage considering the corresponding HPP-untreated sample, at physiological pH, as control following Equation 5, where K_d_ is the hydrophobicity dissociation constant, obtained by plotting the measurement of each dilution vs. the concentration and calculating the slope of the linear regression.
(5)$${\mathrm{Hydrophobicity}}_{\mathrm{Index}\;}\left(\%\right)=\frac{K_{d\;\mathrm{sample}}}{K_{d\;\mathrm{control}}}\cdot 100$$

### 2.8. Total Free Thiol Groups Index

Estimation of the total free thiol groups index (TFT_i_) was carried out following Yongsawatdigul and Park [47]. Briefly: 1 mL of NWCsup was mixed with 9 mL of buffer (dibasic potassic phosphate 0.05 M, urea 8 M and EDTA 10 mM). Then, 4 mL of the diluted NWCsup was incubated at 40 °C for 25 min with 0.4 mL of DTNB 0.1% and the absorbance at 412 nm was measured with a Varioskan^®^ Flash (Thermo Fisher Scientific, Vantaa, Finland). A calibration curve was created by recording the response of standard BSA solutions in the range 0–8 mg/mL. The results were expressed in EU (equivalent units) of BSA (mg/mL), and then normalized considering the corresponding HPP-untreated sample, at physiological pH, as control (Equation (6)).
(6)$${\mathrm{TFT}}_{i}\left(\%\right)=\frac{{\mathrm{TFT}}_{I}{}_{\;\mathrm{sample}}}{{\mathrm{TFT}}_{I}{}_{\;\mathrm{control}}}\cdot 100$$

### 2.9. SDS-PAGE

NWC_sup_ (0.5 mL) were desalted on PD MiniTrap Sephadex G-25 columns (GE Healthcare UK Limited, Buckinghamshire, UK), following the manufacturer instructions.

The electrophoretic separation of WPs was performed under both reducing and non-reducing conditions, using the Agilent Protein 80 Kit on an Agilent 2100 Bioanalyzer equipped with 2100 Expert Software (Agilent Technologies, Inc., Santa Clara, CA, USA), according to the instructions of the provider.

### 2.10. Statistics

All the analyses were performed in triplicate. In the second set of experiments, the effects of independent variables on the dependent variables were checked by ANOVA and Tukey’s Honest Significant Difference (HSD) test, with a statistical significance set at *p* < 0.05. For the α-La yield, the β-Lg A & β-Lg B precipitation degrees, Hydrophobicity Index and total free thiol groups index, a T-test was performed to demonstrate significant differences between control (untreated) and HPP-treated samples. All statistical analyses were carried out using JMP 16.2.0 software (SAS Institute Inc., Cary, NC, USA).

## 3. Results and Discussion

### 3.1. First Set of Experiments

After the HPP treatment, a precipitate was obtained in all the samples, so only the supernatants (NWC_sup_), containing the soluble proteins, were analyzed. The effects of pressure and processing time on α-La yield (Y_α-La_) and α-La Purification degree (Pur_α-La_) at different temperatures are reported as 3D graphs in Figure 1 (physiological pH), Figure 2 (pH 4.6) and Table 2.

The values of the adjusted coefficients (R^2^) always ranged between 0.86 and 0.99, indicating that the models fit well with the experimental data. These are shown, along with the predictive equations, in Appendix A. According to the lack of fit of the model, for the prediction of α-La yield at both pH conditions, the probability value was *p* > 0.5, indicating an adequate adjustment. For the α-La purification degree, *p* was < 0.5. However, the values of the R^2^ and the root square medium error (RMSE) were acceptable enough to consider the model appropriate (Appendix A).

At physiological pH (Figure 1A), Y_α-La_ rose non-linearly with pressure, reaching a theoretical maximum at 288.42 MPa, 47 s of treatment and 21.26 °C. Increasing processing time had some positive effects on yield, albeit only at very low values of pressure. However, pressures over 300 MPa caused a progressive decrease of yield, especially at the highest processing pressures. Initial sample temperatures between 10 °C and 20 °C seemed to improve, to some extent, the yield of α-La, especially when combined with high pressure values and short processing times. Pur_α-La_ increased non-linearly with both pressure and processing time (Figure 1B), reaching the highest theoretical value at 600 MPa, 433 s, and 39 °C, calculated with the equation obtained from the response surface. Temperature treatment below 23 °C provided less favorable results for this parameter.

When processing acidified samples (pH 4.6, Figure 2A), Y_α-La_ decreased linearly with pressure, while the effects of processing time and temperature were less pronounced. A theoretical maximum was achieved at 145 MPa, 47 s and 20.78 °C (Table 2). Effects of temperature were negligible.

Pur_α-La_ showed a behavior similar to that observed at physiological pH (Figure 2B), increasing with both pressure and processing time. The highest value (88%) was theoretically reached at 600 MPa, 433 s, and 39 °C.

NWC acidification at pH 4.6 enhanced the precipitation of the β-Lg and the Pur_α-La_ when HPP was applied, in agreement with Marciniak et al. [36]. A possible explanation could be that, under acid pH conditions, β-Lg formed octamers, possibly enhancing aggregation of this protein during the HPP treatment [15]. Acidification at pH 4.6 decreased Y_α-La_, if compared to samples at physiological pH, likely promoting coprecipitation between α-La and β-Lg, as described in the case of thermal treatments [48]. Partial α-La denaturation probably contributed, regardless of the initial pH of the sample, to decreased Y_α-La_ after HPP treatments higher than 400 MPa, as suggested by Huppertz et al. [49].

From a general point of view, the observed effects of HPP on β-Lg and α-La in bovine native whey concentrate were in good agreement with previous studies carried out on pure compounds [50], bovine whey and milk [35,51], and WPI [52,53,54], suggesting that high holding times, temperatures and pressures enhance β-Lg precipitation and α-La purification degrees.

Based on the results of this first experiment, a second set of pilot plant experiments was carried out to assess the HPP effects on four different batches of native concentrated cow whey. Initial sample temperature (23 °C) and processing pressure (600 MPa) were chosen as fixed processing conditions because of the good balance between Y_α-La_ and Pur_α-La_, while different holding times (up to 30 min) were applied to evaluate the effects of long processing times.

### 3.2. Second Set of Experiments

#### 3.2.1. Main Process Performance Parameters

In accordance with the results from the first set of experiments, around 80% and 94% (physiological and acidified samples, respectively) of the β-Lg initially found in the four NWCs were precipitated after 2 min of HPP treatment (600 MPa, 23 °C) (Figure 3, Appendix A). A quite complete precipitation of β-Lg variants (Pre_β-LgA_ & Pre_β-LgB_ > 94 %) was observed at both physiological and acidified pH starting from 8 min of HPP treatment, without further significant improvements for longer holding times. Both β-Lg variants showed similar behaviors.

The acidification of native whey concentrate also modified the visual appearance of the samples before and after the HPP treatment, favoring the formation of a more compact precipitate in comparison to the samples at physiological pH. The supernatant, after HPP, in samples processed at physiological pH, exhibited a dark color, which could be related to structural and/or aggregation changes of the whey proteins (Appendix A).

It should be underlined that, while working with native whey concentrate, we observed higher β-Lg precipitation degrees than those described by other authors in bovine milk and whey [35,36,55], suggesting that, under our conditions, the initial high protein concentration of the NWCs could have enhanced β-Lg aggregation during the HPP treatment.

The α-La yield (Y_α-La_) at physiological pH was around 55% with an HPP treatment of 2 min; longer holding times progressively reduced the Y_α-La_, up to 13.5% (Figure 4A, Appendix A).

In acidified NWCs, a 2 m HPP treatment provided significantly lower (*p* < 0.05) Y_α-La_ (42.7%) than the corresponding samples at physiological pH, but the α-La yield did not show any significant further reduction for longer holding times (Figure 4A, Appendix A).

When α-La yield was calculated without considering the volume losses (Appendix A) the values ranged from 103.22 to 29.29% (P-pH) and from 103.88 to 82.29% (pH 4.6). This was in agreement with the results obtained by Marciniak et al. [36], who applied HPP in cheese whey, and comparable to those provided by other technologies for fractionating whey proteins of milk and whey, such as membrane filtration (77%) [56], selective precipitation (75.2%) [22], chelating agents (89%) [57] or selective hydrolysis (44–67.9%) [23,24].

The α-La purification degree (Pur_α-La_) in NWC_sup_ at physiological pH was 36.7% after 2 min of HPP, and increased significantly (*p* < 0.05) with longer holding times, up to 79.3% at 30 min. NWC_sup_ acidification before HPP significantly increased Pur_α-La_, which reached a mean value of 78.1 % with a holding time of 2 min and a maximum of 92% at 30 min (Figure 4B, Appendix A). Marciniak et al. [36] also reported that acidification (pH 4.6) of cheese whey before HPP treatment (15 min, 600 MPa) increased the Pur_α-La_; notwithstanding, working with NWCs at physiological pH, we reached a higher α-La purification degree than those reported by these authors for similar holding times, indicating that processing a concentrated product may increase Pur_α-La_.

Overall, under our conditions, the range of values for Pur_α-La_ was comparable with those reported in the literature for selective precipitation (52–83% [58]), use of chelating agents (90% [57]), membrane separation combined with tryptic hydrolysis (44% [23]) or continuous centrifugal separation after selective precipitation (99.4% [22]). The combination of membrane and high pressure processing could be an alternative to very efficient technologies with a high economical cost (chromatographic separation) or to more economical but less selective ones (filtration), for the obtention of an α-La-enriched fraction.

In general terms, under our conditions, an α-La-enriched fraction with a good balance between Y_α-La_ and Pur_α-La_ could be achieved from acidified NWCs processed at ambient temperatures by applying HPP at 600 MPa for 4 min (α-La yield = 46.16%, α-La purification degree = 80.21%). These conditions were similar to those suggested by Marciniak et al. [36] for cheese whey, once the differences in α-La yield calculation are considered.

However, in cases in which it would be preferable to preserve the native proteins (e.g., to be used as ingredients in infant food formulations), an HPP treatment of NWCs at physiological pH and ambient temperature at 600 MPa for 4 min could still provide a Pur_α-La_ of 43.97% with a Y_α-La_ of 49.13% (Appendix A).

Acidification before HPP (pH 4.6) promoted the aggregation of β-Lg induced by high-pressure, providing higher α-La yield and purification with very short HPP treatments, compared to those observed working with NWCs at physiological pH.

#### 3.2.2. Total Free Thiol Groups Index (TFT_i_)

TFT_i_ can provide information about the tertiary and quaternary conformation of proteins. An increase of the TFT_i_ value may indicate that the protein suffered a structural change that exposed the free sulfhydryl groups, while a decrease may suggest their oxidation or the formation of new disulfide bonds [59,60]. TFT_i_ increased after acidification in untreated NWCs, indicating that acidification itself could affect the quantity and structure of free sulfhydryl groups (Figure 5). These effects of pH differences have been mentioned by Marciniak et al. [36], who stated that acidification of whey (pH 4.6) increased fluorescence intensity, indicating a change in protein conformation.

However, under our conditions, the TFT_i_ was significantly reduced (*p* < 0.05) for both NWC_sup_ at physiological and acidic pH after HPP treatment (600 MPa, 23 °C) for all holding times.

At physiological pH, the decrease of TFT_i_ was less pronounced for short holding times (up to 4 min). Meanwhile, in the case of acidified samples, TFT_i_ was already virtually zero after an HPP treatment of 2 min. These differences could be explained by the modest effect of the HPP on the α-La free sulfhydryl groups at physiological, pH as suggested by Rodiles-López et al. [61].

In any case, it seemed that HPP could promote both the oxidation of the available free thiol groups and/or the formation of new disulfide bonds in NWCs, as previously observed in acidified whey treated with dynamic high-pressure homogenization [62] and in pure bovine β-Lg [63].

#### 3.2.3. Hydrophobicity Index

Surface hydrophobicity is related to the number of hydrophobic groups exposed on the surface of the protein in contact with an aqueous polar environment. The increase of the hydrophobicity index has been associated to structural changes in the proteins’ surface, leading to the exposure of nonpolar amino acids, while its decrease suggests the formation of protein aggregates by hydrophobic interactions [62,64,65].

A significant increase of the hydrophobicity index was observed in NWC_sup_ treated with HPP at physiological pH with a holding time of 2 and 4 min, in comparison with untreated samples (Figure 6). Other authors observed similar results [62,66,67] after HPP treatments of commercial whey protein concentrates and isolates at physiological pH between 300–800 MPa.

However, HPP holding times higher than 15 min provoked a significant decrease (*p* < 0.05) of the hydrophobicity index in NWC_sup_ at physiological pH, if compared to the untreated samples. Liu et al. [64] also reported a reduction in hydrophobicity index in commercial whey protein concentrate (WPC) HPP treated at 600 MPa with holding times higher than 5 min. On the other hand, the HPP treatments always induced a significant reduction of the surface hydrophobicity in acidified NWC_sup_, in comparison with the untreated samples, even for short holding times (2 min). Thus, pressure and NWCs’ acidification seemed to have a similar effect on the hydrophobic groups, which, once exposed on the surface of the proteins, could react and promote the formation of protein aggregates.

#### 3.2.4. SDS-PAGE

Figure 7 shows the native SDS-PAGE profiles of untreated (controls) and HPP-treated (2, 4 and 15 min; physiological pH and 4.6) NWC_sup_ samples. Monomeric forms of α-La and β-Lg corresponded to the bands at 12.3 KDa and at 18.4 KDa, respectively.

The electrophoretic profiles confirmed the progressive reduction of the β-Lg bands with increased HPP holding times, the synergic effects of the NWCs’ acidification, and the higher baro-resistance of α-La, in accordance with our HPLC analysis results (Appendix A) and the literature [36,49,68].

Liu et al. [64], Bouaouina et al. [66] and Marciniak et al. [36] described the presence of protein aggregates after treating with HPP WPC, WPI and cheese whey, respectively. However, in our study, there were not clearly visible differences between the SDS-PAGE under reducing and non-reducing conditions, suggesting that whey concentration could promote the formation of protein aggregates—which were precipitated and eliminated during the post-processing steps (centrifugation), as indicated by Marciniak et al. [69]. Additionally, the 3.5 KDa band, appearing in all NWCs treated with HPP, could have been related to peptide formation, due to the partial degradation of whey proteins over time. Further studies should be carried out to support this observation.

## 4. Conclusions

This study provided evidence of native whey protein behavior following HPP treatments, confirming the higher baroresistance of α-La, compared to β-Lg, in native whey concentrate under different treatment conditions (pH, pressure, temperature, and processing time).

The combination of sample acidification (pH 4.6) and pressure (600 MPa) provided acceptable α-La yield and α-La purification degree values with a short HPP treatment (4 min) at ambient temperature (23 °C). However, further characterization of the α-La-enriched fraction would be necessary, since sample acidification and HPP seemed to have a synergic effect on the protein structure, as suggested by the lower values of both total free thiol groups index and hydrophobicity index observed in the NWC_sup_ from acidified samples (if compared to those processed at physiological pH).

Under our working conditions, the general effects of HPP on α-La and β-Lg agreed with results reported in other studies on cow milk or whey. Notwithstanding, our results also indicated that the use of native whey concentrate could improve the precipitation degree of β-Lg, in comparison to raw milk or whey.

This effect was more evident when working with native whey concentrate at physiological pH, when we reached higher α-La purification degree than those reported by other authors under comparable HPP conditions. The results suggested that the initial high protein concentration in NWCs could have enhanced the β-Lg aggregation during the HPP treatment.

Additional experiments at the pilot plant level are planned to optimize, with a decanter, the separation of the α-La-enriched fraction from bovine NWCs after the HPP treatment, and to provide more information about the functionality of the α-La-enriched fraction. Furthermore, the applicability of the combined process (membrane concentration and HPP treatment) will be further assessed to valorize cheese whey from dairy industry or to obtain α-La-enriched fractions from whey of other animal species.

## Figures and Tables

**Figure 1 foods-12-00480-f001:**
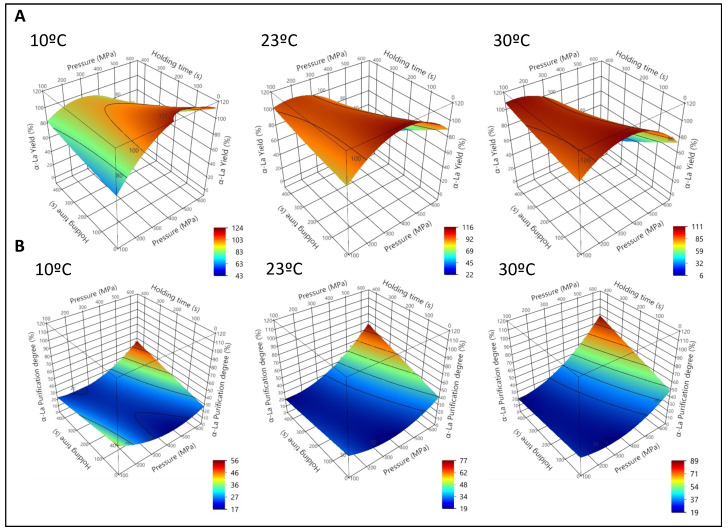
First experiment. Native whey concentrate supernatant (NWC_sup_) at physiological pH. Representation of the surface responses for α-Lactalbumin yield (**A**) and α-La purification degree (**B**) after high pressure processing (HPP) treatments as a function of pressure (P), holding time (t) and initial sample temperature (10 °C, 23 °C and 30 °C), according to the equations from Appendix A.

**Figure 2 foods-12-00480-f002:**
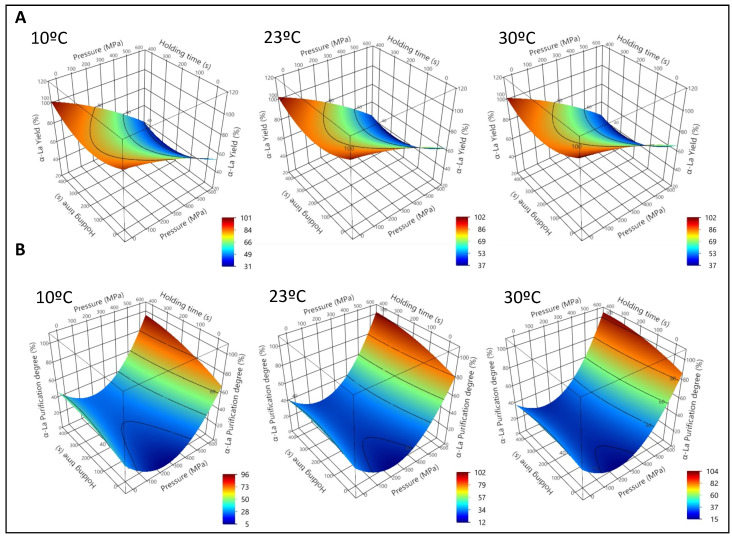
First experiment. Acidified (pH 4.6) native whey concentrate supernatant (NWC_sup_). Representation of the surface responses for α-Lactalbumin yield (**A**) and α-La purification degree (**B**) after high-pressure processing (HPP) treatments as a function of pressure (P), holding time (t) and initial sample temperature (10 °C, 23 °C and 30 °C), according to the equations from Appendix A.

**Figure 3 foods-12-00480-f003:**
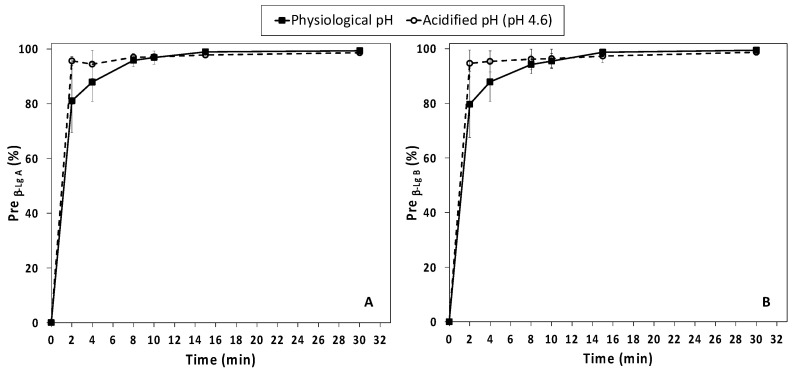
Second experiment. Precipitation degree of β-Lactoglobulin (β-Lg) (**A**,**B**) variants (Pre_β-LgA_ and Pre_β-LgB_) as a function of HPP treatment time (min) for native whey concentrate supernatant (NWC_sup_) at physiological pH and acidified at pH 4.6. (Mean values ± standard deviation, *n* = 4).

**Figure 4 foods-12-00480-f004:**
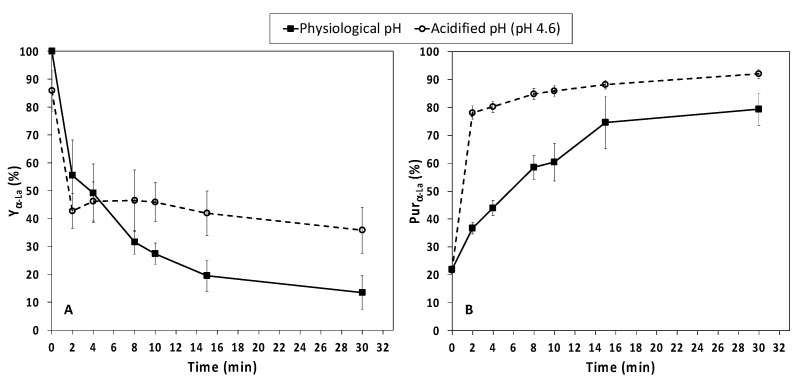
Second experiment. α-La Yield (**A**) and α-La purification degree (**B**) as a function of high-pressure processing (HPP) treatment time (min) for native whey concentrate supernatant (NWC_sup_) at physiological pH and acidified at pH 4.6. (Mean values ± standard deviation, *n* = 4).

**Figure 5 foods-12-00480-f005:**
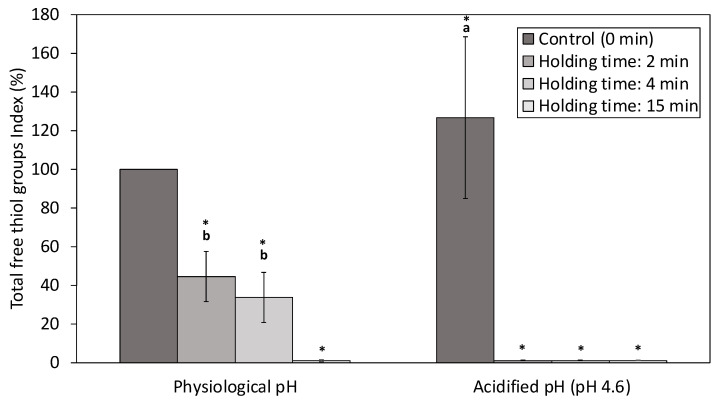
Second experiment. Total free thiol groups index of native whey concentrate supernatant (NWC_sup_) after HPP treatments (0, 2, 4 and 15 min; 600 MPa; 23 °C) of NWCsup, at physiological pH and acidified at pH 4.6. (Mean values ± standard deviation, *n* = 4). a,b indicate significant differences between means (*p* < 0.05) according to Tukey test. * Significant differences with untreated samples.

**Figure 6 foods-12-00480-f006:**
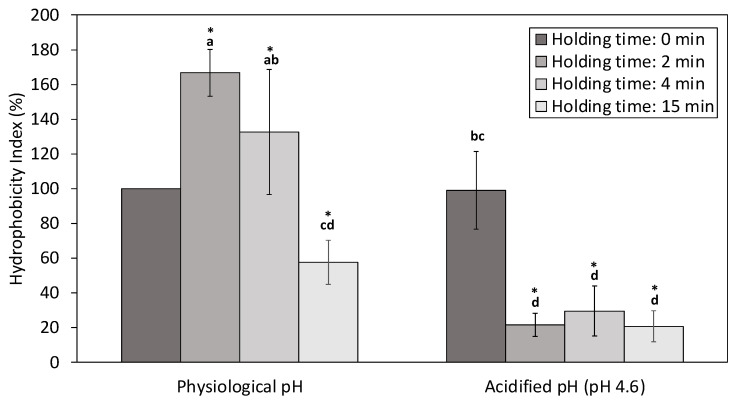
Surface hydrophobicity results. Hydrophobicity Index (%) of high-pressure processing (HPP) treated native whey concentrate supernatant (NWC_sup_) (2, 4 and 15 min) regarding the control. Bar errors mean standard deviation. a–d Mean values with different superscripts differ significantly (*p* < 0.05) according to Tukey test. * Significant differences with untreated samples.

**Figure 7 foods-12-00480-f007:**
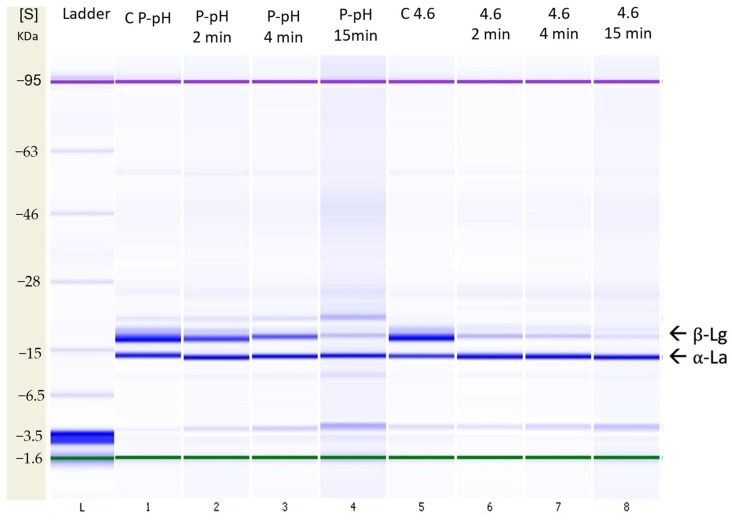
Non-reducing SDS-PAGE for control (C) and high-pressure (600 MPa) treated samples (2, 4 and 15 min) for native whey concentrate supernatant (NWC_sup_) at physiological pH (P-pH) and pH 4.6. α-La = α-lactalbumin; β-Lg = β-lactoglobulin.

**Table 1 foods-12-00480-t001:** Composition of the five native whey concentrate (NWC) batches before treatment. Std. dev. = standard deviation.

	NWC0	NWC1	NWC2	NWC3	NWC4	Mean ± Std. dev.
Ash (% *w*/*w*)	0.74	0.76	0.77	0.79	0.82	0.78 ± 0.03
Total dry matter (% *w*/*w*)	19.60	18.56	19.49	20.82	24.04	20.50 ± 2.13
Lactose (% *w*/*w*)	5.33	5.13	4.84	4.83	4.93	5.01 ± 0.21
Fat (% *w*/*w*)	0.05	0.03	0.05	0.04	0.08	0.05 ± 0.02
Protein (% *w*/*w*)	13.48	12.64	13.83	15.20	18.67	14.76 ± 2.37
pH	6.55	6.44	6.53	6.56	6.64	6.54 ± 0.07
Concentration factor *	28.68	23.40	24.69	26.67	33.95	27.48 ± 4.13

* Concentration factor calculated as final protein concentration after ultrafiltration, divided by the initial protein concentration before ultrafiltration.

**Table 2 foods-12-00480-t002:** First set of experiments. Protein concentrations and main process performance parameters of α-Lactalbumin (α-La)-enriched fraction after high-pressure processing (HPP) (supernatant). * Processed 600 MPa for technical reasons. P = pressure; T = initial sample temperature.

			Physiological pH					Acidified pH (4.6)				
P (MPa)	T (°C)	Time (s)	α-La Concentration (mg/mL)	β-LgB Concentration (mg/mL)	β-LgA Concentration (mg/mL)	α-La Yield (%)	α-La Purification Degree (%)	α-La Concentration (mg/mL)	β-LgB Concentration (mg/mL)	β-LgA Concentration (mg/mL)	α-La Yield (%)	α-La Purification Degree (%)
Control			52.62	45.23	132.44	-	22.85	50.09	43.20	126.62	78.16	22.78
145	23	240	51.83	44.67	130.16	99.02	22.87	52.14	45.01	130.85	75.91	22.87
200	10	90	49.84	43.06	125.52	93.74	22.82	52.81	45.67	132.94	78.46	22.82
200	10	390	48.85	40.39	123.09	91.87	23.01	50.27	43.55	126.42	77.26	22.83
200	35	90	53.91	46.16	134.97	97.74	22.94	51.19	44.19	129.21	79.10	22.79
200	35	390	55.01	46.89	137.19	104.72	23.01	55.02	47.25	138.45	85.57	22.86
400	7	240	47.99	40.73	111.76	93.75	23.94	53.18	25.81	78.53	58.97	33.76
400	23	47	56.23	44.53	129.73	109.11	24.39	53.48	37.37	100.19	73.63	27.99
400	23	240	51.56	35.12	103.82	97.66	27.07	54.85	29.07	69.18	62.72	35.83
400	23	240	53.45	35.13	102.19	101.14	28.02	57.99	31.19	69.57	63.54	36.53
400	23	240	58.76	38.98	114.46	114.98	27.69	53.18	28.67	63.63	57.31	36.56
400	23	240	57.30	37.61	109.85	108.05	27.98	55.93	29.89	66.33	58.24	36.77
400	23	240	50.54	32.41	94.75	91.29	28.44	50.98	27.36	59.75	49.46	36.92
400	23	433	58.84	29.98	89.86	86.55	32.93	62.00	29.86	59.55	57.89	40.95
400	39	240	55.78	24.09	71.00	84.72	36.97	50.21	25.70	48.36	52.28	40.41
600	10	90	55.99	36.36	108.26	108.46	27.91	52.04	10.08	20.89	36.89	62.69
600	10	390	57.19	20.17	61.32	70.51	41.24	50.15	3.91	9.33	42.60	79.11
600	23	240	52.83	11.71	36.67	51.43	52.19	51.04	2.97	6.28	41.25	84.66
600	35	90	51.82	12.03	37.21	62.78	51.28	51.00	3.00	6.83	54.95	83.83
655 *	35	390	27.99	1.55	4.37	21.37	82.52	40.53	1.45	3.26	52.18	89.59

## Data Availability

Data unavailable for privacy reasons.

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
