# Peer review of "Separation of α-Lactalbumin Enriched Fraction from Bovine Native Whey Concentrate by Combining Membrane and High-Pressure Processing"

_foods, 2023, doi:10.3390/foods12030480_

Round 1

Reviewer 1 Report

This work shows the effect of HPP treatments on main proteins (α-La and α-Lg) in native concentrated bovine whey, obtained from fresh milk by ultrafil-85 tration at pilot plant level without any heat treatment. Overall, the introduction part is good and the experimental data are detailed. Some issues:

(1) In the title, the authors emphasized “combining membrane and HPP”, but in the introduction part the authors did not introduce the membrane.

(2)  If the authors could provide further characterizations of the α-La enriched fraction after HPP?More evidences should be given to show the changes of protein structure and its functionality.

(3) If the authors could simply and quantitatively compare the economic efficiency and selectivity of the HPP-membrane method with other approaches?

Author Response

In the title, the authors emphasized “combining membrane and HPP”, but in the introduction part the authors did not introduce the membrane.

A short introduction to membrane technology has been added (L85).

 If the authors could provide further characterizations of the α-La enriched fraction after HPP?More evidences should be given to show the changes of protein structure and its functionality.

We agree that further characterizations of the α-La enriched fraction after HPP is needed. We did not include it in this first set of experiments, but we will consider specific studies during the optimization of the process at pilot plant level (L713).

 If the authors could simply and quantitatively compare the economic efficiency and selectivity of the HPP-membrane method with other approaches?

A comparison has been added (L504-507). Detailed quantitative economic comparison was considered out of the scope of our work due to the lack of available published data.

Reviewer 2 Report

In line 144, the expression "identify optimal HPP conditions", requires a mathematical procedure specified for this purpose. Specify the expression.

Present a table with the predictive regression models coefficients and remove them from the titles of the figures. Insert the lack of fit of the model.

Present a discussion or explain on the use and process conditions of membranes in the study.

It is necessary to improve the resolution of the figures.

Present in Table 1 the standard deviation of the values.

How was the handling of the five batches of native concentrated whey (NCW) during the experiments?

Present the experimental design for the second set of experiments. Specify the application conditions of the central composite design. In general, detail the statistical management of the study.

Author Response

In line 144, the expression “identify optimal HPP conditions”, requires a mathematical procedure specified for this purpose. Specify the expression.

We did not apply a mathematical procedure. The sentence has been rephrased (L192)

 Present a table with the predictive regression models coefficients and remove them from the titles of the figures. Insert the lack of fit of the model.

Done; Table S1 summarizes the predictive regression models coefficients and the lack of fit of the models.

 Present a discussion or explain on the use and process conditions of membranes in the study.

All membrane processes were performed according to the fabricant specifications, a sentence has been added in Materials and Methods (L113).

 It is necessary to improve the resolution of the figures.

Done, figure resolution has been improved.

 Present in Table 1 the standard deviation of the values.

Mean and std. dev. of the 5 whey batches have been added in Table 1.

 How was the handling of the five batches of native concentrated whey (NCW) during the experiments?

Details about whey handling have been added (L131, L132)

 Present the experimental design for the second set of experiments. Specify the application conditions of the central composite design. In general, detail the statistical management of the study.

No central composite design was applied in the second set of experiments. Description of the experiments has been modified to improve the clarity of the presentation (L203-L207; L310)

Reviewer 3 Report

Review Report:

 Separation of α-Lactalbumin Enriched Fraction from Native Concentrated Bovine Whey by Combining Membrane and High-Pressure Processing

 Main Impression

The study and knowledge adressed in this manuscript is important to the field of valorization of whey. The work is an important contribution to developers of infant formulas, as these results can improve the yield of important protein fractions. The article fits into the scope of the journal, with its focus on different aspects of food technology, processing and application. Yet, more documentation is needed to reach the industrial relevance.

However, using the phrase Native Concentrated Whey is slightly misleading. It is better rephrased to Native Whey Concentrate, so that the word native explains the quality of the whey, rather than the concentration process. Also change the abbreviation NCW to NWC, which is also used in other publications about concentrated native whey (e.g. in ref.11).

Overall, the language in the manuscript is good and easy to read, but some phrases can lead to misunderstanding. These are pointed out above and in the detailed report. Detailed comments and suggestions for improvement are given below.

 ABSTRACT

The abstract neatly gives the key to the article.

 INTRODUCTION

The introduction gives a good background and knowledge base on the productions of proteinfractions from whey. The introduction also clearly shows the need for more detailed information on the processing and application of native whey.

L35 replace dot with a comma, prior to “as a bioactive…”

L57 Please correct the references given for hydrolysis to [23,24]

 MATERIALS AND METHODS

L117 Please correct misspelled word “ultrafiltration”

L129-135 To my experience modifications to reference methods are often needed. Here there are none? Is that correct? Why are the ISO standards not listed in the reference list?

L138-139 What is meant by “the come-up time”? Is it the same as the compression rate during HPP?

L147-148 what is the practical relevance of the times and temperatures selected? Maximum pressure given in table 2 is 600 MPa, here it is written 655 MPa. What is correct?

L152 replace the word “aleatory” with the word “random”

L157-158 what is meant by the sentence? Are the samples treated independently from eachother? One sample at a time? Please rephrase.

L163-164 Please rewrite. What is a blinding, and what is meant by relative codification?

L166-175 Is the method of HPLC quantification developed by the authors, or does the method have a reference?

 RESULTS AND DISCUSSION

Nb! From page 8, the numbers of lines and pages are reset. The following comments are referring to this new numbering.

L23 The abbreviation HHP appears with no explanation. Is this referring to the “high hydrostatic pressure” in the reference given, or is it a mis-typing of HPP? The abbreviation HHP appears more places in the paper. Please correct.

L24 insert “of” after the word “because” in the sentence.

L30 “cow whey” should be replaced or rephrased. For example it can be written “whey from cow’s milk”

L75 HHP?

L92 Please explain the abbreviation “WPs”.

L119-120 What is meant by “double the Purα-La”? Compared to what? Please rephrase the sentence to avoid misunderstanding.

L145 The reference is not nr. 56, but nr. 57. Please correct.

L177 HHP?

L194-195 Can the peptide and it’s origin be checked/identified? How?

 CONCLUSION

The conclusion neatly sums up the findings of this work, and points out further investigations. The importance of using native whey as starting material for HPP purification of α-Lactalbumin, is pointed out.

L199-200 “treatment duration” can be replaced with “time”

L219-220 The sentence indicate the similar research should be performed on cheese whey? What about continuing the work on Native Whey?

 TABLES AND FIGURES

Throughout the article the figures and tables are referred to in bold letters. Please make sure that this is consistent.

Table 2 Titles of column 2 and 3 must change place. There is an asteriks for “processed 600 MPa for technical reasons” – where is the asteriks in the table?

Figure 1 Please use the same units in the caption as in the figure. (MPa not P, and s not t)

Figure 2 See comment above

Figure 7 Please add KDa in the title of the first column

Table S1 No need for second footnote. Please remove

Figure S1 Are the colours in the print correct? Why is the colour of native whey so dark? Can it be explained by the changes induced to proteins in this work?

Author Response

Main Impression

It is better rephrased to Native Whey Concentrate, so that the word native explains the quality of the whey, rather than the concentration process. Also change the abbreviation NCW to NWC, which is also used in other publications about concentrated native whey (e.g. in ref.11).

Done; suggested changes applied throughout the manuscript.

INTRODUCTION

L35 replace dot with a comma, prior to “as a bioactive…”

Done

 L57 Please correct the references given for hydrolysis to [23,24]

Done

MATERIALS AND METHODS

L117 Please correct misspelled word “ultrafiltration”

Done

 L129-135 To my experience modifications to reference methods are often needed. Here there are none? Is that correct? Why are the ISO standards not listed in the reference list? 

Minor modifications included in the text (L176). The ISO protocols are now listed in the reference list.

 L138-139 What is meant by “the come-up time”? Is it the same as the compression rate during HPP?

Sentence was rephrased by indicating compression rate (L186).

 L147-148 what is the practical relevance of the times and temperatures selected? Maximum pressure given in table 2 is 600 MPa, here it is written 655 MPa. What is correct?

The maximum pressure value from the experimental design was 655 MPa, but for the technical limitations of our HPP equipment, those samples were processed at 600 MPa (this is now better indicated in Table 2),

L152 replace the word “aleatory” with the word “random”

Done

L157-158 what is meant by the sentence? Are the samples treated independently from each other? One sample at a time? Please rephrase.

This sentence was rephrased to improve clarity (L207)

L163-164 Please rewrite. What is a blinding, and what is meant by relative codification?

This sentence was rephrased to improve clarity (L213)

 L166-175 Is the method of HPLC quantification developed by the authors, or does the method have a reference?

A reference was added.

 RESULTS AND DISCUSSION

Nb! From page 8, the numbers of lines and pages are reset. The following comments are referring to this new numbering.

Numbering has been revised.

 L23 The abbreviation HHP appears with no explanation. Is this referring to the “high hydrostatic pressure” in the reference given, or is it a mis-typing of HPP? The abbreviation HHP appears more places in the paper. Please correct.

Abbreviation has been standardized throughout the manuscript.

 L24 insert “of” after the word “because” in the sentence.

Done

 L30 “cow whey” should be replaced or rephrased. For example it can be written “whey from cow’s milk”

Done, rephrased throughout the manuscript.

 L75 HHP?

Abbreviation for High Pressure Processing (HPP) has been standardized throughout the manuscript.

 L92 Please explain the abbreviation “WPs”.

WPs defined in introduction (L32)

 L119-120 What is meant by “double the Purα-La”? Compared to what? Please rephrase the sentence to avoid misunderstanding.

This sentence has been rephrased to improve clarity of presentation.

 L145 The reference is not nr. 56, but nr. 57. Please correct.

Done

 L177 HHP?

Abbreviation for High Pressure Processing (HPP) has been standardized throughout the manuscript.

 L194-195 Can the peptide and it’s origin be checked/identified? How?

The possible formation of peptides in HPP treated samples was not further investigated because it should require analytical technologies not included in this study, but we agree that it should be an interesting topic. A sentence has been added (L653)    

CONCLUSION

L199-200 “treatment duration” can be replaced with “time”

Done; replacement applied throughout the manuscript.

 L219-220 The sentence indicate the similar research should be performed on cheese whey? What about continuing the work on Native Whey?

Additional information has been added to indicate the interest on scaling up the process for bovine native whey and to study to explore the applicability of the process for other animal sources (L713-717).

TABLES AND FIGURES

Throughout the article the figures and tables are referred to in bold letters. Please make sure that this is consistent. 

Done; change applied throughout the manuscript.

 Table 2 Titles of column 2 and 3 must change place. There is an asterisk for “processed 600 MPa for technical reasons” – where is the asterisk in the table?

Table has been modified and the meaning of the asterisk explained in the legend.

 Figure 1 Please use the same units in the caption as in the figure. (MPa not P, and s not t)

As suggested by Reviewer #1 the predictive coefficients have been moved to Table S1. Meaning of abbreviations P, T and t are explained in the table legend.

 Figure 2 See comment above

As suggested by Reviewer #1 the predictive coefficients have been moved to Table S1. Meaning of abbreviations P, T and t are explained in the table legend.

 Figure 7 Please add KDa in the title of the first column

Done

 Table S1 No need for second footnote. Please remove

Former Table S1 has been renamed Table S2. Table footnote has been rephrased to improve clarity.

 Figure S1 Are the colours in the print correct? Why is the colour of native whey so dark? Can it be explained by the changes induced to proteins in this work?

We agree that the darker color could be due to the structural changes provoked by the combination of whey concentration and HPP, but no specific studied were carried out. A sentence has been added in the manuscript (L453).

Reviewer 4 Report

Dear authors,

the work is very well structured

Regards

Author Response

Thank you very much for your comments.